# Atomic Simulations of (8,0)CNT-Graphene by SCC-DFTB Algorithm

**DOI:** 10.3390/nano12081361

**Published:** 2022-04-15

**Authors:** Lina Wei, Lin Zhang

**Affiliations:** 1Key Laboratory for Anisotropy and Texture of Materials, Ministry of Education, Northeastern University, Shenyang 110819, China; 1610119@stu.neu.edu.cn; 2School of Electrical Additionally, Information Engineering, Ningxia Institute of Science and Technology, Shizuishan 753000, China; 3Department of Materials Physics and Chemistry, School of Materials Science and Engineering, Northeastern University, Shenyang 110819, China

**Keywords:** DFTB, carbon nanotube, graphene, atomic simulation

## Abstract

Self-consistent density functional tight binding (SCC-DFTB) approaches were used to study optimized structures, energy, differential charge density, and Mülliken populations for the (8,0) carbon nanotubes (CNTs) connected to the graphene having different topology defects. Based on the calculations, nine seamless (8,0)CNT-graphenes were selected. For these connected systems, geometric configurations of the graphene and nanotubes were characterized, and the nearest neighbor length of C-C atoms and average length were obtained. The intrinsic energy, energy gap, and chemical potential were analyzed, and they presented apparent differences for different connection modes. Differential charge densities of these connection modes were analyzed to present covalent bonds between the atoms. We have also thoroughly analyzed the Mülliken charge transfer among the C atoms at the junctions.

## 1. Introduction

Low dimensional carbon nanostructured materials have shown outstanding performance in various fields. Carbon nanotube (CNT)-graphene composites are composed of two carbon allotropes, concerning environmentally friendly carbon elements, which are used as a remarkable enhancer to improve plasma frequency [1], microwave adsorption [2], and EMI shielding effect [3]. The applications of the CNT-graphene composites are also particularly involved in device sensing, energy, supercapacitors, wearable devices, and other flexible electronic fields owing to their great feasibility [4,5]. Here, the regulation approach based on graphene provides a new idea for improving the performance of graphene-based micro-supercapacitors [6], and the multi-void graphene joint with 3D CNTs is also a three-dimensional conductive network to promote thermal and electric conduction [7]. In the CNT-graphene composites, pillared graphene paper-based supercapacitors exhibit excellent electrochemical performances and cyclic stabilities [8,9,10,11]. As a counterpart of 3D carbon architectures, the pillared graphene nanostructure consists of graphene planes pillared with nanotube fragments, and it provides augments of hydrogen storage [12,13,14,15].

In laboratory, Li. et al. detected a unique seamlessly bonded CNT-graphene hybrid nanostructure introduced in an interlayer for efficient and stable perovskite solar cells, and the power conversion efficiency improved significantly [16]. Maarouf et al. compounded a CNT-graphene hybrid material with a resistance considerably lower than neat graphene [17]. Ali et al. fabricated and characterized CNT-graphene composite-based piezoresistive pressure samples, providing a potential conduction mechanism [18]. Peng et al. studied and proposed a biomimetic material enhancement strategy by using CNT to enhance 3D graphene electrodes [19]. Park et al. reported CNTF-Cu-Gr wires obtained by introducing graphene into Cu electroplated on CNTFs having high performances [20]. Theoretically, Baowan et al. mathematically predicted 16 different defects that graphene might link to CNTs [21]. Novaes et al. reported that there are only three symmetrical modes for these CNT-graphene connections through the first-principle calculations [22]. Moreover, some studies reported that the defects on the graphene (sheet) and nanotube (pillar) parts play an important role in yielding magnetism or thermal transportation of the 3D pillared graphene [23,24,25,26]. Previous studies proved stability of topological defects in graphene including dislocation quadropole and dipole arrangements, stacking faults, partial dislocations, and grain boundaries [27,28]. Nguyen et al. reported a series of 12 transition metals supported on various graphene models with graphitic nitrogen defects [29]. Shyam et al. characterized with electronic and magnetic properties of in-plane defect motifs in the graphene by a first-principles study from [30]. Jeong et al. found that the structure of the dislocation defect with two 5–7 pairs becomes more stable than a local haeckelite structure, which is composed of defect units of three pentagons and three heptagons (555–777 defect) when the number of vacancy units is ten and over. Scanning tunneling microscopy (STM) reveals that the 5–7 pair defects perturb the wave functions of electrons near Fermi level to produce a superlattice pattern [31].

In the two tubes, the zigzag (8,0) tube has a small chemical hardness, and the carbon atoms in the cross section of the (8,0) tube are axisymmetric when it is adjacent to the graphene atoms. Therefore, one interesting issue arises naturally to understand connection patterns in the (8,0)CNT-graphene. The study of the low dimensional nanomaterials for CNT-graphene is looking forward to a matter worthy of expectation to build devices having good binding and stability at the nanoscale. Accounting for the fact that electrical information determined by experiment is hardly possible for these connectors, computer simulations based on semi-empirical quantum mechanics, such as density functional tight binding (DFTB), are particularly well-suited to characterize microscopic details at atomic scale [32,33,34,35,36,37], whereas the system involving hundreds of atoms cannot be solved by ab initio calculation.

In the present work, the structures and electrical properties are investigated by DFTB simulations for these connecting modes of the graphenes, respectively, having 16 defects linked to the (8,0) CNTs. These values are examined including geometric configurations, energy analysis, electron difference density, and Mülliken charge. The atomic simulations contribute to identify the defect’s role in the connection between the carbon nanotube and graphene as well as the bonding between the dangling atoms. The structural and electronic related information of carbon nanotube-graphene heterojunctions will be helpful in experimental characterization of the interface at atomic scale for novel 3D nanostructures based on carbon [38].

## 2. Simulation Methodology

The present simulations used DFTB+ software, which is developed by Bremen University in Germany [39]. Within the SCC-DFTB formalism, total energy of the system in the method is given by the following formula [40,41].
(1)$$E_{scc}=\sum_{i}n_{i}\langle \phi_{i}\left|{\hat{H}}^{0}\right|\phi_{i}\rangle +\frac{1}{2}\sum_{\alpha \beta }\Delta q_{\alpha }\Delta q_{\beta }\gamma_{\alpha \beta }+E_{rep},$$
where $n_{i}$ is occupancy of molecular orbit $\phi_{i}$, ${\hat{H}}^{0}$ is Kohn–Sham operator. The Mülliken charge $\Delta q_{\alpha }$ is calculated from the Mülliken population of the α atom $q_{\alpha }$ and the number of valence electrons for the neutral atom $q_{\alpha }^{0}$ ($\Delta q_{\alpha }\;$= $q_{\alpha }^{0}$−$q_{\alpha }$) [42]. $E_{rep}\;$is the two-body repulsive potential. $\gamma_{\alpha \beta }$ is defined as follows:(2)$$\gamma_{\alpha \beta }=\left(1/r_{\alpha \beta }\right)-S_{\alpha \beta }$$

Here, $r_{\alpha \beta }$ is the distance between two atoms and $S_{\alpha \beta }$ is the short-range correction term among atomic nuclei.

Mülliken populations were calculated from Linear Combination of Atomic Orbitals (LCAO) coefficients [42,43,44]. For one system containing *N_atom_* atoms (A, B… are used as symbols of these atoms) and *N* electrons, molecular orbital is given below:(3)$$\phi_{i}=\underset{A}{\sum }\underset{\mu }{\sum }cA_{\mu i}\chi A_{\mu }$$
where $\chi A_{\mu }$ is the atomic orbital of atom *A*, and *c* orbit coefficient. Then,
(4)$$\phi_{i}^{*}\phi_{i}=\underset{A}{\sum }\underset{B}{\sum }\underset{\mu }{\sum }cA_{\mu i}cB_{\lambda i}\chi A_{\mu }\chi B_{\lambda }$$

Integrating both sides of this equation, and multiplied by *n_i_* to get the following,
(5)$$n_{i}=n_{i}\underset{A}{\sum }\overset{A}{\underset{\mu }{\sum }}c_{A\mu i}^{2}+2n_{i}\underset{A>B}{\sum }\sum \overset{A}{\underset{\mu }{\sum }}\overset{B}{\underset{\lambda }{\sum }}cA_{\mu i}cB_{\lambda i}S_{A\mu ,B\lambda }$$

This equation can be used to analyze the charge distribution. The charge distributed on the atomic orbital is
(6)$$n\left(i,A\mu \right)=n_{i}c_{A\mu i}^{2}$$

We sum the *i*, and the charge on the atomic orbital $\chi A_{\mu }$ is
(7)$$n\left(A\mu \right)=\underset{i}{\sum }n(i,A_{\mu })=\underset{i}{\sum }n_{i}c_{A\mu i}^{2}$$

By summing all the atomic orbitals of atom *A*, the charge of atom *A* is obtained:(8)$$n\left(A\right)=\underset{\mu }{\sum }n\left(A\mu \right)$$

The total overlapping charge of the μ orbital of *A* atom and the λ orbital of *B* atom is
(9)$$n\left(i,A\mu ,B\lambda \right)=2n_{i}c_{A\mu i}c_{B\lambda i}S_{A\mu ,B\lambda }$$

The constructed (8,0) SWNTs having 96 carbon atoms were performed geometry optimization, where the C-C bond length was 1.420 Å being parallel to the tube axis and 1.434 Å being non-parallel to the tube axis, and the diameter was 6.337 Å. Meanwhile, the initial graphene was 18 Å in length and width. We started by removing the carbon atoms from the graphene to form the 16 possible defect patterns as given in the literature [21]. Sixteen possible graphene defects used to link carbon nanotubes (8,0) have been shown in Figure 1. Here, this figure and the following Figure 2, Figure 3 and Figure 4 are viewed from Weblab code. In Figure 2, the defect #1 of the graphene was marked counterclockwise with the carbon atom numbers “1-8”, and the atoms at the bottom of the carbon nanotube are also marked with numbers “1-8”. The top dangling bonds of C atoms in the (8,0) CNT were saturated with a ring of hydrogen atoms. During structural relaxation, the carbon atoms at the top of the carbon nanotube have the configurations with those at their tube geometries owing to these H atoms. We connected the “1” atom of the graphene to the “1” atom at the bottom of the nanotube, and then “2-2”, “3-3”, “4-4”, “5-5”, “6-6”, “7-7”, and “8-8”. Therefore, (8,0)CNT-graphenes were established by the #1. The other corresponding connection configurations of the (8,0)CNT-graphenes were obtained in the same way. Geometries were optimized by using the conjugate gradient method. In the simulation cells containing these CNT-graphene structures, periodical conditions are applied along the three directions of these cells.

(8,0)CNT-graphene positions are described as the following. The maximum x-coordinate of carbon atoms in the graphene is Xmax and the minimum x-coordinate is Xmin. Xmax–Xmin is divided equally into intervals with an increment of ∆x. There are ∆N_X_ carbon atoms in the i (I = 1,2,3…) interval [X_min_ + (i − 1)∆x, X_min_ + i∆x]. Therefore, we define F_x_ as follows: F_x_ = ∆N_x_/N(10)

Here, it should be noted that N is the number of carbon atoms in the graphene for the (8,0)CNT-graphene. Similarly, the maximum y-coordinate of carbon atoms in the graphene is Ymax and the minimum y-coordinate is Ymin. Ymax–Ymin is divided equally into intervals with an increment of ∆y. There are ∆N_y_ carbon atoms in the i (I = 1,2,3…) interval [Y_min_ + (i − 1)∆y, Y_min_ + i∆y]. Therefore, we define F*_y_* = ∆N_y_/N. The maximum z-coordinate of carbon atoms in the graphene is Zmax and the minimum z-coordinate is Zmin. Zmax–Zmin is divided equally into intervals with an increment of ∆z. There are ∆N_z_ carbon atoms in the i (I = 1,2,3…) interval [Z_min_ + (i − 1)∆z, Z_min_ + i∆z]. Therefore, we define F*_z_* = ∆N_z_/N.

To characterize the interaction of these atoms, the formula for the intrinsic energy per atom of these modes was calculated as follows:(11)$$E_{ins}\left(CNT\text{-}graphene\right)=\left[E_{tot}\left(CNT\text{-}graphene\right)-n_{C}E\left(C\right)-n_{H}E\left(H\right)\right]/N$$

Here, *E_ins_*(*CNT-graphene*) is intrinsic energy per atom, *n_c_* is the number of carbon atom, and *n_H_* is the number of hydrogen atom, and *N* is the total number of atoms. *E(C)* and *E(H)* are the monatomic energies of carbon and hydrogen, respectively. *E_tot_* (*CNT-graphene*) is the total energy of CNT-graphene.

In general, there is an energy difference between the highest occupied molecular orbital (HOMO) energy E_HOMO_ and the lowest unoccupied molecular orbital (LUMO) energy E_LUMO_. The energy gap is given by *Eg = E_LUMO_* − *E_HOMO_*.

## 3. Results and Discussion

### 3.1. Structure and Energy Analysis

Figure 3 shows the optimized structures of sixteen graphenes having defects after structural relaxation. With the exception of #7, all other graphenes show obvious ups and downs along the z direction. In the x-y planes, some graphenes cannot keep their initial configurations. There are connections between neighboring atoms for the #5, #11, #14, #15, and #16 defective graphenes, which are marked by black ellipses in Figure 3. Table 1 lists these graphenes and (8,0)CNT-graphenes. Here, “Y” indicates that the graphene can hold its defective configurations as given in Figure 1, and the seamless connection between the atoms of the tube and graphene, whereas “N” means failure. Except for the five graphenes that cannot keep the origin defect configurations, the seamless connection modes cannot appear in the connection of tube and #4 and #12 graphenes.

Figure 4a shows the nine geometries of the seamless (8,0)CNT-graphene, where the first row is the top view of the x-y plane for the nine structures, the second is the side view of the x-z plane, and the third the y-z plane. As shown in this figure, after the connection, these graphenes show more significant shape changes than before, where there are apparent deviations along the X and Y directions owing to the pulling and pressing from the connected carbon nanotubes. Different from the other eight modes, the graphene of the #1 mode has obvious fold patterns in the x-y plane.

As shown Figure 4b for the configuration of sample #4, there are individually suspended atoms in the connecting region between the (8,0)CNT and grapheme. The similar phenomenon can be also found in the other failed configurations including #11, #12, #14, and #15 CNT-graphene couple modes. In addition, both the coupling modes #5 and #16 form cage-like closed CNT ports.

In order to quantitatively analyze the geometries of the (8,0)CNT-graphenes, we calculated fraction of the carbon atoms for the CNT-graphenes along three directions as given by the Equation (11). As illustrated in Figure 5a of the deviation along the x direction for the atoms in graphene, five connections modes of #3, #9, #6, #8 and #1 have peak values near 0. The two modes of #2 and #13 achieve the peak values at 0.2 and 0.4, respectively. Both #7 and #10 modes have the peak value at −0.1. The deviations suggest that most of the tubes are upright above the graphene along the x direction, and the tube of the #13 has the largest inclination angle in the negative x direction. For the case along the y direction, most of the tubes present apparent tilt toward y positive direction, whereas #2 and #10 negative direction. Figure 5c illustrates that the fluctuations along z direction also show obvious differences for these connection modes. The obvious undulation of graphene surface comes from the carbon tube on it. For the initial configurations, the angle between the carbon nanotube axis and *x*-axis or *y*-axis is 90 degrees, while the angle between the carbon nanotube axis and *z*-axis is 0 degree. For the nine seamless connection modes, the angle between the CNT tube and coordinate axis minus that corresponding to the initial position is the change values as illustrated in Figure 5d. As indicated the black line of the angle between tube axis and *x*-axis, the maximum positive value of the mode #13 corresponds to the positive maximum deflection of the graphene carbon atom along the x-axis. The #13 also has a large positive value between the tube axis and *z*-axis. The large positive angle between the tube axis and *y*-axis or *z*-axis for the #8 suggests that the tube of this connection also shows obvious inclination. The small angles for the #9 indicate that the tube is vertical on the graphene. The change of the angle between the tube axis and the coordinate axis for these carbon nanotubes is related to the deflection of carbon atoms in the graphene. In the meantime, the asymmetry of bond-length in the graphene is amplified near the tube, thus increasing the deflection density in these regions.

In the following Table 2, we measured the length of eight C-C atomic pairs linked at the connection points. For the two cases of connected upright, the #9 is the largest average bond length, whereas the #3 the smallest one. For the #1 and #2, 4-4 atoms have the same bond length of 1.389 Å, while the other bond length is greater than 1.4 Å. It can be noted that the largest length of the #2 is apparently larger than that of the #1, suggesting that the inclination angle between the carbon tube and graphene in the #2 is larger than that in the #1. Figure 5c shows that the axis of the carbon nanotube in the #2 is far away from the *z*-axis and close to the positive direction of the *y*-axis compared with that in the #1, and the angles with the *y*-axis and *z*-axis increases. For the #6, the length of four atomic pairs is 1.432 Å, the other four 1.414 Å. For the other connection modes, the scatter distribution of bond lengths suggests that there are also inclination angles between the tube and graphene. Especially for the #13 having the largest inclination angle, there is a larger length of four pairs and smaller length of the other four pairs.

For inorganic crystals composed of carbon or silicon, our previous works [33,34,35,36,37] show that when selecting appropriate Slater–Koster tables, the DFTB calculations can give similar results to those from the first principle calculations within DFT formalism. Figure 6a shows the intrinsic energy of nine connection modes in these seamless (8,0)CNT-graphenes. It can be found that the #3 has the maximum value of 6.3954 eV, whereas the #1 mode has a relatively low energy. The intrinsic energy reflects the bonding strength among the atoms. It can be found that the differences for these connection modes are only the second digits after the decimal point, suggesting the strong bonding in the seamless modes. The energy gap is obtained by subtracting the energy of the highest occupied molecular orbital (HOMO) from that of the lowest unoccupied molecular orbital (LUMO).The two energies can be obtained from the detailed out file after performing the DFTB calculations. The energy gap is plotted in Figure 6b, where the values of the gap are in all the cases near zero, indicating that these CNT-graphenes present metallicity. The energy gap both #3 and #13 modes are larger than those of the other ones. Among the nine modes, the #13 mode has the maximum energy gap of 0.0050 eV, while #8 coupling mode has the minimum energy gap of 0.0001 eV. As illustrated in Figure 6c, the #1, #6 and #8 have relatively high chemical potentials of about −4.50 eV, and the #2, #3, and #7 have a value of −4.57 eV. The #13 has the minimum value of −4.62 eV, indicating that its chemical stability is the highest. Here, the zero chemical potential comes from Fermi level energy at 0 K.

### 3.2. Differential Charge Density and Mülliken Population

Figure 7 shows differential charge densities of these seamless modes along three axes. These images are from VMD software [45]. The differential charge density at one site in space is obtained from the total charge density of the simulated system minus the product between atomic density and the ratio of total charges of the system to those of isolate atoms. For these modes, the blue represents negative density around the position of one C atom, and red positive density between the C atoms corresponding to the covalent bond. As shown in this figure, there are two blue fold stripes along the x-axis in the middle of the graphene for the #1, suggesting that there are some C atoms in the strips that obviously protrude from the surface of the graphene owing to the pulling and pressing from the tube’s atoms. As the graphenes of the #9 and #13 show only small bulges, the densities are more uniform than those of the others. 

Figure 8 shows the Mülliken population both maximum and minimum of graphene and CNT in the (8,0)CNT-graphene. Among these modes, the minimum Mülliken population of the carbon nanotubes in the CNT-graphene are significantly smaller than that of the graphenes, indicating that lost charge of the CNTs are much more than those of the graphenes. For the #1, #3, #6, or #13 mode, the maximum Mülliken population of the carbon nanotube in the CNT-graphene is greater than that of the graphene. Getting charge of the CNTs are much more than those of the graphenes in the four modes.

As shown in Figure 9, Mülliken charges of the carbon atoms around the connection knots from the graphene and CNT present apparent differences, where there are twenty-two carbon atoms involved in the graphene [21] and sixteen carbon atoms in the carbon nanotubes. In this figure, the small ball represents a CNT atom at the knots, and the large ball an atom of the graphene. For the #1 and #2 modes, the amount of charge transfer near the junction is mostly similar, but the difference is that the two defect positions marked as “7” and “8” in the graphene. The angle between 7-7, 8-8 and tube axis in the structure of #2 is much smaller than that of #1, so the gain and loss charge of the #2 mode is more than that of the #1. For the #3 case, the atoms that get and lose charge are arranged symmetrically along the x and y axes. Ten of the sixteen atoms in the tube lose charge and six get. Among these ten atoms in which the charge is obtained, eight atoms are directly connected with the carbon atoms of the graphene. Eight of the 22 atoms in the graphene connected to the carbon nanotube get charges, and the other 14 atoms lose. Considering the atoms in the #6 and #7 modes, most of the charge transfers are very similar, except for the two atomic defect positions marked as “2” and “3” in the graphene. The angle between 2-2, 3-3 and tube axis in the #7 is much smaller than that of the #6, and amount of charge obtained and lost in the #7 mode are more than that in the #6. For the #8 and #9, they are both symmetrical around the *x*-axis. If there are not the two defect positions marked as “7” and “8” in the graphene, the #8 would be symmetrical just like the #9. The amount of charge obtained and lost among the carbon atoms at the junction of 7-7 and 8-8 for the #8 are higher than those for the #9. The carbon atoms near the “2-2” and “3-3” connections in the #10 have more gain and loss of charge than those near the other connections. With an eye to the #13, the defect arrangement of the graphene is symmetrical, resulting in the symmetrical arrangements of the atoms getting and losing charge at the junctions. Three of the sixteen atoms in the carbon nanotube gain charge and thirteen lose. The atoms in the graphene are dominated by charge loss.

## 4. Conclusions

The geometric structural optimization and electronic information of the (8,0)CNT-graphene are investigated by using SCC-DFTB simulations at atomic scale. (8,0)CNT-graphene is a suitable choice for the carbon nanotubes linked with grapheme having topological defects within the present DFTB formalism. Carbon nanotubes having a larger diameter need to be connected with graphene having larger size topological defects, and this cannot be performed through even the semi-empirical DFTB calculations. Geometrically, thinner nanotubes can hardly form a seamless connection with graphene having topological defects. In the present work, the hydrogenation for the top C atoms of the nanotubes is used to represent their infinite length. For these 16 topology defects constructed in the graphene, there are only nine seamless connection modes with the (8,0) CNTs. After connection, the graphene shows obvious fluctuation along the *z*-axis, and carbon tubes take forms mainly including upright and inclined. The phenomena can be also found in 3D graphene-CNT-Ni heteronanostructures. Here, the heteronanostructures embedded in electrodes and the ionic membrane of the artificial muscles present significantly bending deformation. Some (8,0)CNT-graphene systems with vertical tubes have relatively high intrinsic energy and chemical stability. Owing to the pulling and pressure from the connected carbon tubes, the morphologies of the graphene present apparent differences, which greatly affect the distribution of electrical densities of the graphene. The locally morphological differences are similar to the localized part of experimentally growing CNTs on graphene, where optimized interfacial bonding state play a great significance in the strength/toughness and EMI shielding effectiveness. From the Mülliken population differences, it can be found that the lost amount of the charge on the tube’ atoms are higher than that the graphene’s ones. The transfer of Mülliken charge obviously occurs among the atoms at the junction between the tube and graphene, implying that there are weak ionic bonds between these atoms in addition to strong covalent bonds. The present calculations provide us the possible structures and charge distributions as well as the energy information for coupling CNTs with grapheme, and improve our understanding for such systems on electrical level. These will be helpful in constructing novel carbon nanotube arrays on 2D materials with excellent thermal properties.

## Figures and Tables

**Figure 1 nanomaterials-12-01361-f001:**
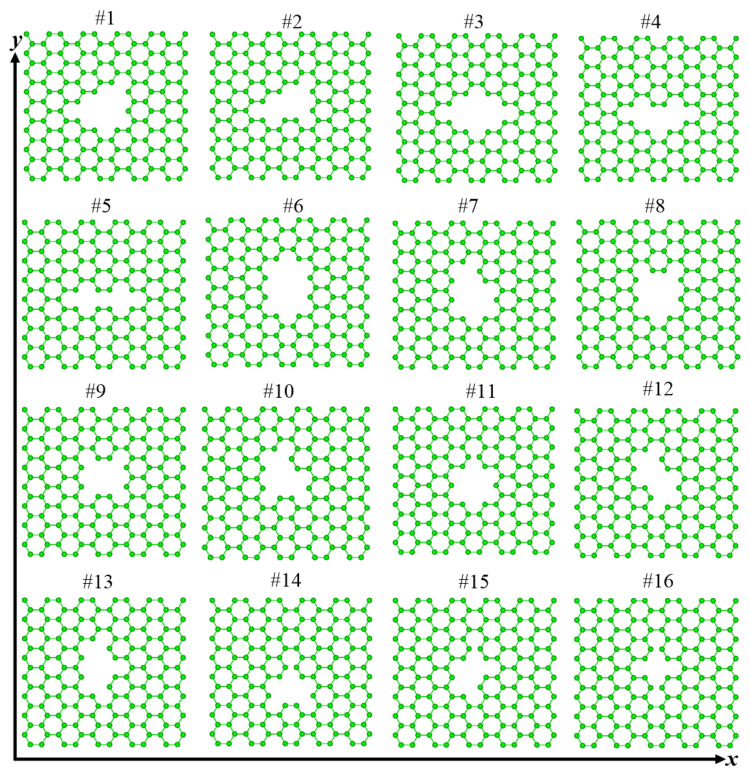
Sixteen initial structures of graphene having defects.

**Figure 2 nanomaterials-12-01361-f002:**
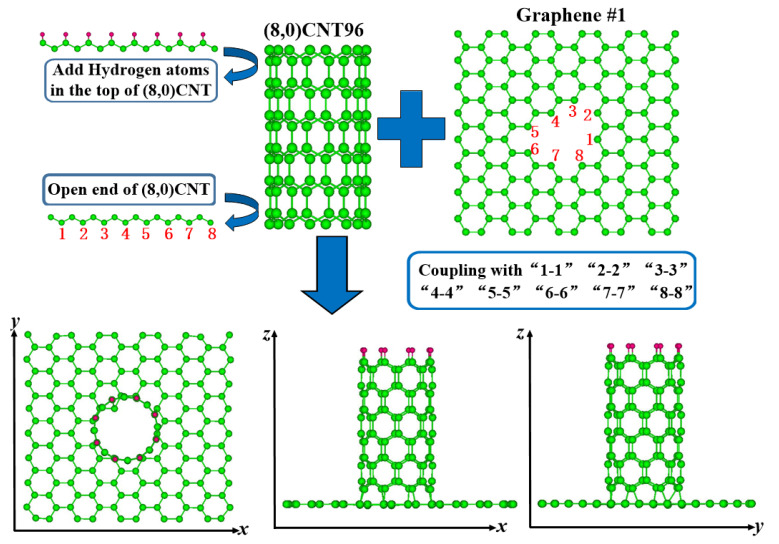
Schematic diagrams of initially constructed (8,0)CNT-graphene configurations #1 structures.

**Figure 3 nanomaterials-12-01361-f003:**
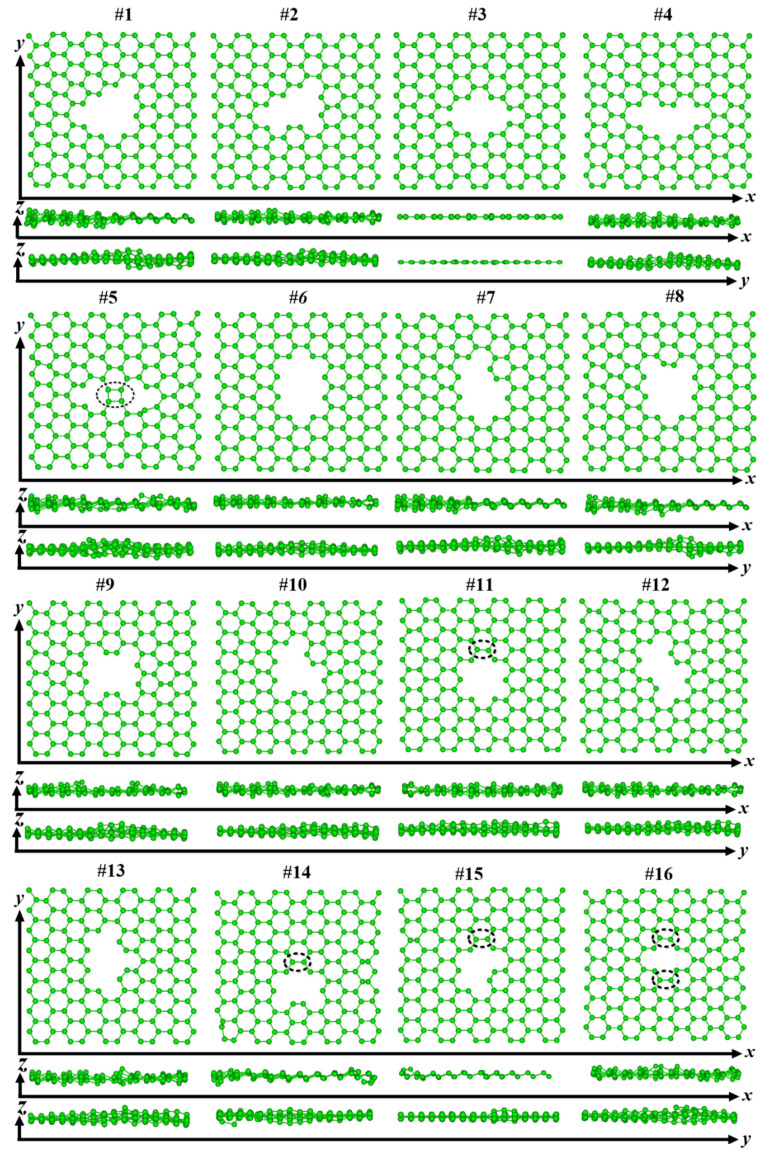
Optimized sixteen possible structures of the grapheme having defects.

**Figure 4 nanomaterials-12-01361-f004:**
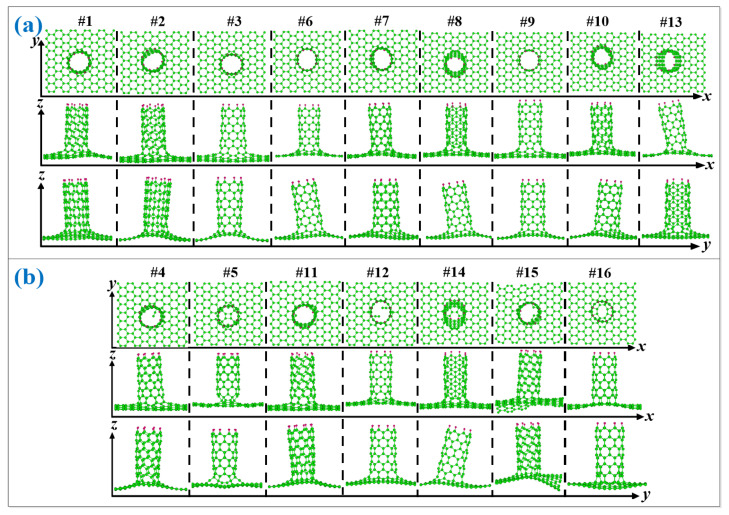
(**a**) Optimized nine seamless structures of (8,0)CNT-graphene for configurations of #1,#2,#3,#6,#7,#8,#9,#10,#13. (**b**) Optimized seven failed structures of (8,0)CNT96-graphene for configurations of #4,#5,#11,#12,#14,#15,#16.

**Figure 5 nanomaterials-12-01361-f005:**
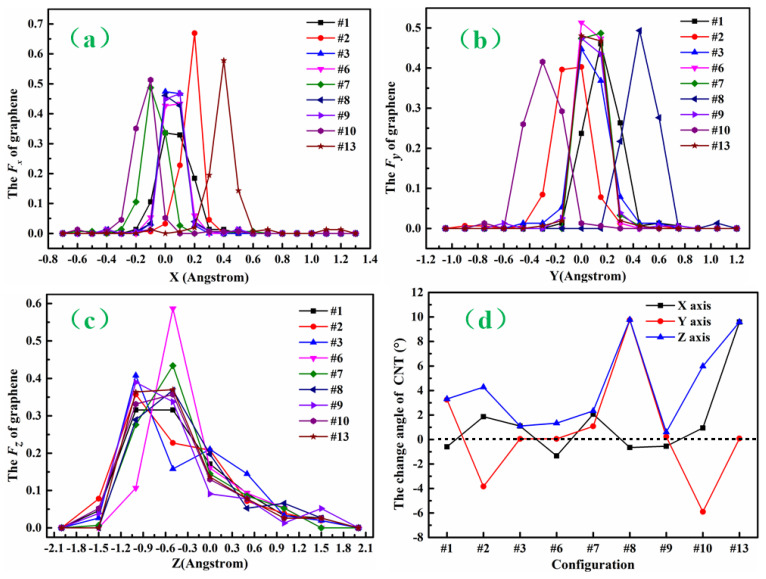
The statistics of position change in the (8,0)CNT-graphene for seamless #1, #2, #3, #6, #7, #8, #9, #10, and #13. (**a**) The f_x_ function broken line; (**b**) The f_y_ function broken line; (**c**) The f_z_ function broken line; (**d**) The change angle of CNT.

**Figure 6 nanomaterials-12-01361-f006:**
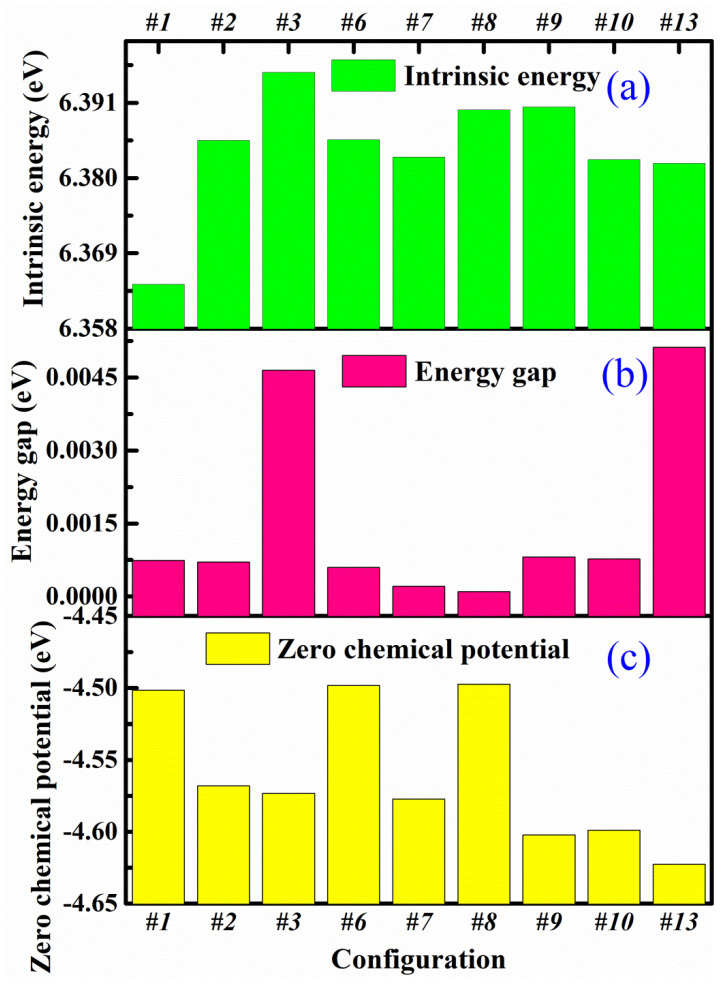
(**a**) Intrinsic energy. (**b**) Energy gap. (**c**) Chemical potential of (8,0)CNT-graphene.

**Figure 7 nanomaterials-12-01361-f007:**
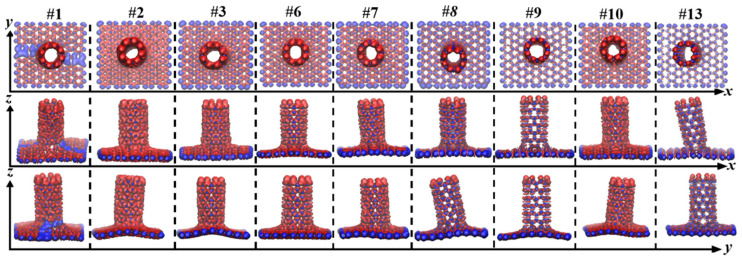
Differential charge density of (8,0)CNT96-graphene for configurations of #1, #2, #3, #6, #7, #8, #9, #10, and #13.

**Figure 8 nanomaterials-12-01361-f008:**
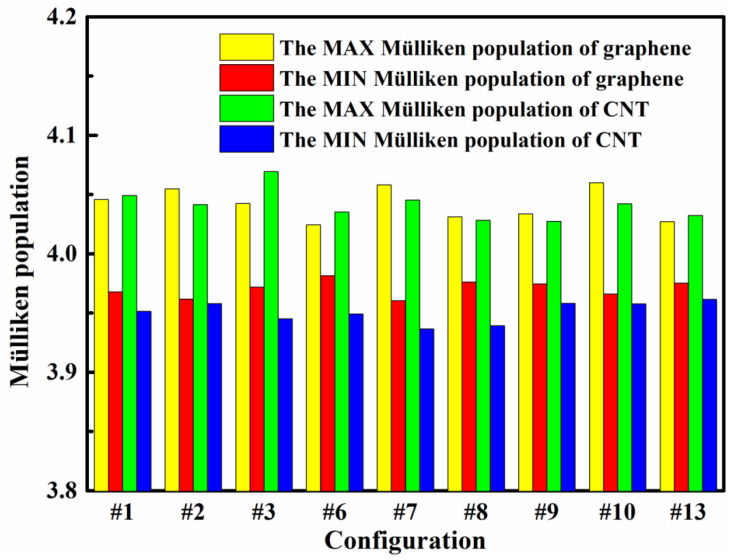
The Mülliken population both maximum and minimum of graphene and CNT in the (8,0)CNT-graphene.

**Figure 9 nanomaterials-12-01361-f009:**
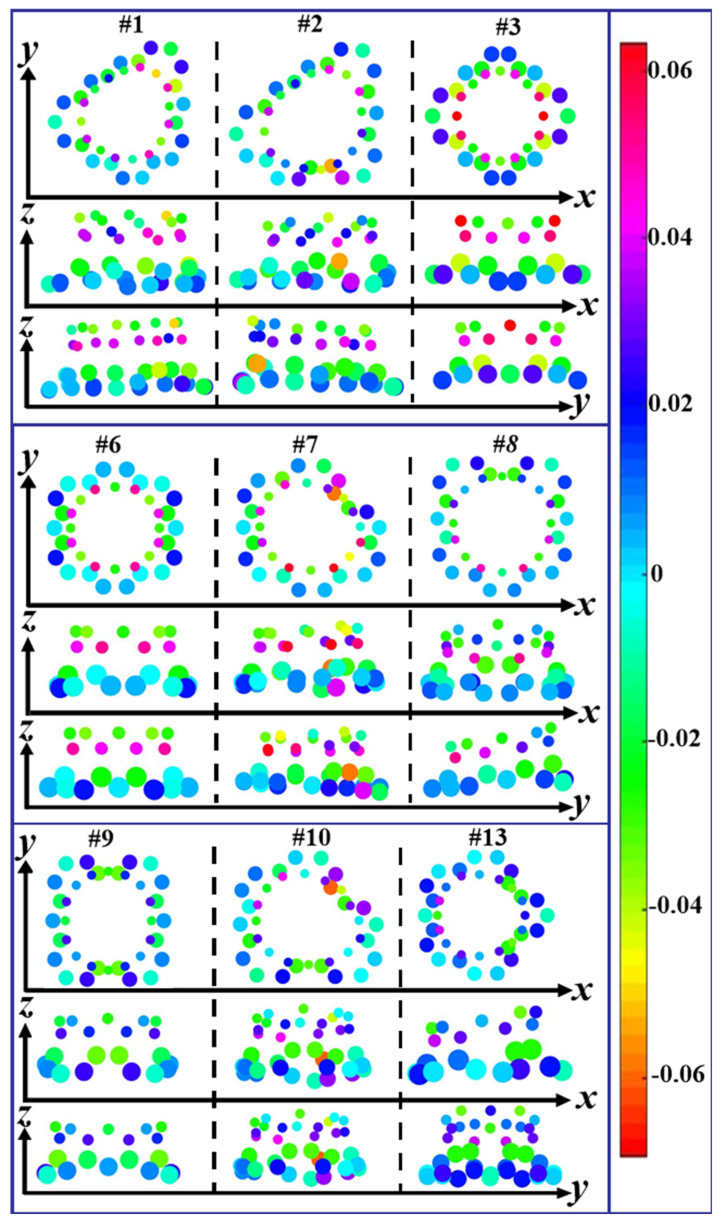
The Mülliken charge near around the connection knots of (8,0)CNT96-graphene by these seamless links.

**Table 1 nanomaterials-12-01361-t001:** Statistical results for 16 kinds of defects for independent graphene and (8,0)CNT-graphene.

Defects	1	2	3	4	5	6	7	8	9	10	11	12	13	14	15	16
graphene	Y	Y	Y	Y	N	Y	Y	Y	Y	Y	N	Y	Y	N	N	N
(8,0)CNT-graphene	Y	Y	Y	N	N	Y	Y	Y	Y	Y	N	N	Y	N	N	N

**Table 2 nanomaterials-12-01361-t002:** The C-C bond length around the connection joints of (8,0)CNT-graphene according to the numbering order in the references [21].

Configuration	C-C Bond Length/Å								
Connection	1-1	2-2	3-3	4-4	5-5	6-6	7-7	8-8	Average
#1	1.403	1.405	1.444	1.389	1.441	1.411	1.413	1.431	1.417
#2	1.408	1.418	1.439	1.389	1.437	1.434	1.435	1.473	1.429
#3	1.417	1.417	1.414	1.413	1.415	1.415	1.413	1.414	1.415
#6	1.414	1.414	1.432	1.432	1.414	1.414	1.432	1.432	1.423
#7	1.409	1.439	1.476	1.408	1.406	1.407	1.425	1.412	1.423
#8	1.410	1.407	1.472	1.472	1.407	1.410	1.416	1.416	1.426
#9	1.410	1.410	1.468	1.468	1.410	1.410	1.468	1.468	1.439
#10	1.401	1.443	1.471	1.415	1.410	1.405	1.468	1.474	1.436
#13	1.455	1.455	1.462	1.382	1.397	1.396	1.383	1.462	1.424

## Data Availability

Not applicable.

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
