# Peer review of "Atomic Simulations of (8,0)CNT-Graphene by SCC-DFTB Algorithm"

_nanomaterials, 2022, doi:10.3390/nano12081361_

Round 1

Reviewer 1 Report

The paper presents research on atomic simulations of the (8,0) carbon nanotubes connected to the graphene having different topology defects by SCC-DFTB algorithm. The presentation of methods and scientific results in the current form is satisfactory for publication in the Nanomaterials journal. The minor and significant drawbacks to be addressed can be specified as follows:
1.    Lines 39, 42, and 44 (see also all the manuscript). (i) Problem with citing papers. Would you please remove the names and leave only the last names? For example, Ahmed A. Maarouf ---> Maarouf. (ii) et al ---> et al. (dot after al).
2.    Introduction. No information about “pillared graphene”.
Google research – graphics – keywords: “pillared graphene”.
Froudakis’ work is available.
Google research – graphics – keywords: “pillared graphene Froudakis”.
3.    Introduction. Many works on defects on/in “pillared graphene” were omitted.
Google research – graphics – keywords: “pillared graphene defects”.
4.    Page 5, Fig. 1. Have periodic conditions been taken into account for these structures? If not, please answer the following question: What would be the impact of considering periodic conditions on stabilising the graphene-like fragment (Figs. 3 and 4?
5.    Page 9, Tab. 1, table captions. What does “Y” and “N” mean?
6.    Line 179. Equ. ---> Eq.
7.    Lines 71 – 76. The goals were not clearly stated.
8.    Lines 74 – 76. “The data from atomic simulations contribute to the experimental characterization at atomic scale.” What experimental works do the presented results confirm?
9.    Line 236. What program was used to prepare Figs. 1-4? VMD?
10.    Line 245. Link?
11.    Fig. 8. Mulliken Population ---> Mulliken population.
12.    Conclusions. No reference to the experiment. 

Author Response

Dear Reviewer :

Thank you very much for sending us the constructive report of our manuscript (manuscript code Nanomaterials-1351761). We have followed all suggestions by the reviewer on a point-to-point basis and hope that the paper thereby can be accepted for publication in NANOMATERIALS. The changes that we have made are marked by red in the revised manuscript. In detail, We have made a detailed description in the attachment.

We hope this paper is suitable for “Nanomaterials” and deeply appreciate your consideration of our manuscript. If you have any queries, please don’t hesitate to contact me at the address below.

Thank you and best regards.

Yours sincerely,

Lina Wei

Name: Lin Zhang

E-mail: zhanglin@imp.neu.edu.cn

Reviewer 2 Report

I find the topic of this work very interesting and the authors use the potentiality of DFTB, which other DFT methods cannot provide. However, although the results seem to be good, the presentation of them is very poor. My main concerns:

  • In the introduction, an unnecessary  detailed explanation of DFTB is given, which can be found in the literature. Important references about the stability of defects on graphene are not mentioned.
  • The definitions of Fx, Fy and Fz are not clear: which are the values of Δx, Δy and Δz? At which value of y and z is calculated Fx? At which value of x and z is calculated Fy? At which value of x and y is calculated Fz?
  • Fig.5: Which are the units of X, Y and Z?
  • I guess, the gap is obtained from the eigenvec.out file from DFTB, but it is not mentioned...
  • The values of the gap is in all the cases near zero, the apparently larger values for #3 and #13 than those of the other ones are not relevant
  • The differences in the intrinsic energy are also not relevant, they differ by the second digits after the decimal point.
  • zero chemical potential: the authors mean the Fermi level? Their differences are also not significant.
  • How are the differential charge densities obtained? Which are the magnitudes of these densities? no scale is shown in Fig.7...
  • The Link of reference (21) does not work

Author Response

(The authors gave the same response as above.)

Reviewer 3 Report

In the submitted work the authors studied the structures and some basic electronic properties of a number of "hybrid" structures built from graphene sheets and carbon nanotubes. Such systems are well known for their potential in hydrogen storage in their bulk phase

The authors considered sixteen different types of graphene defects and used a method of a relatively low computational cost (fully justifiable due to the size of the systems) to conduct their study and draw their conclusions. As it shown only 9 out of the 16 graphene defects chosen can produce a seamless fusion of the studied nanotube. This is also one of most important conclusions of this work.

This work might be publishable after the authors address the following issues

1. Unfortunately, the main conclusion of the work cannot be easily generalized because the authors have chosen to conduct their study using only one nanotube type. From my point of view this shortcoming limits the applicability of the results. Therefore, the authors should either give strong arguments within the discussion that will provide a convincing answer to this above question or extend their study using other type of nanotubes having different circumference sizes. For the last choice, given the fact that the systems are quite large, the choice to consider nanotubes of smaller lengths would be fully justified. If I am not mistaken the length of the CNT should not be of major importance for these slabs

  1. The bandgaps presented are extremely weak. How accurate is the chosen approach for the reliable determination of such small energetic differences between the conduction and the valence bands? The authors could verify their results by performing some reference computations on the relaxed DFTB+ structures using well established DFT methods.
  2. The structures of the non-seamless “failed” configuration should be included either in the MS or as supporting material. This would be very useful for the readers

Author Response

(The authors gave the same response as above.)

Reviewer 4 Report

The manuscript addresses the atomistic simulation of CNT/graphene junctions. A series of junction configurations is investigated, and the corresponding energetics and charges are predicted. I have two major issues with this paper. First, it is very unclear why this research is important, and what the practical implications are of the results. The results of the research are merely stated, but I don’t understand why the results are important, or how they relate directly to a specific engineering application. The conclusions section is silent on this issue, as is the results and discussion section. In other words, “so what?”. Second, the paper does not include any background information on the previous work done on computational modeling of CNT/graphene junctions (see, for example, some papers from Vikas Varshney). It is unclear how the previous generation of research on this subject relates to this paper.

Author Response

(The authors gave the same response as above.)

Round 2

Reviewer 1 Report

The authors have made a substantial improvement for this article. The manuscript can be accepted for publishment in the present form.

Author Response

Dear Reviewer:

Thank you for your recognition.

Thanks a lot and kind regards.

Yours sincerely,

Lina Wei

Name: Lin Zhang

mail:zhanglin@imp.neu.edu.cn

Reviewer 2 Report

The authors did not answer to all my concerns...

"•    In the introduction, an unnecessary  detailed explanation of DFTB is given, which can be found in the literature. Important references about the stability of defects on graphene are not mentioned."

The introduction has not been shorted in the technicals details.

The two references introduced (23-24) are not related to the stability  of vacancies. There are experimental evidences (for example Scanning transmission electron microscopy) in the literature about this important point, for example, it has been shown that defects tend to heal spontaneously by filling up with either nonhexagon, graphene-like, or
perfect hexagon 2D structures.

"    •    The definitions of Fx, Fy and Fz are not clear: which are the values of Δx, Δy and Δz? At which value of y and z is calculated Fx? At which value of x and z is calculated Fy? At which value of x and y is calculated Fz?"

This point has not been answered...

Author Response

Dear Reviewers :

Thanks a lot for sending us the constructive report of our manuscript (manuscript code Nanomaterials-1351761). We have followed all suggestions by the reviewer on a point-to-point basis and hope that the paper thereby can be accepted for publication in NANOMATERIALS. The changes that we have made are marked by red in the revised manuscript. In detail,please see the attachment.

We hope this paper is suitable for “Nanomaterials” and deeply appreciate your consideration of our manuscript. If you have any queries, please don’t hesitate to contact me at the address below.

Thanks a lot and kind regards.

Yours sincerely,

Lina Wei

Name: Lin Zhang

E-mail:zhanglin@imp.neu.edu.cn

Reviewer 4 Report

The authors have satisfactorily addressed my comments.

Author Response

(The authors gave the same response as above.)
